



**Historic carbon burial spike in an Amazon floodplain lake linked to riparian**
**deforestation near Santarem, Brazil**
Luciana M. Sanders[1], Kathryn Taffs[1], Debra Stokes[3], Christian J. Sanders[2], Alex Enrich-
Prast[4,5], Leonardo Nogueira Amora[6,7], Humberto Marotta[6,7]
[1]*Southern Cross Geoscience, Southern Cross University, P.O. Box 157, Lismore, NSW 2480, Australia.*
[2]*National Marine Science Centre, School of Environment, Science and Engineering, Southern Cross*
*University, Coffs Harbour, New South Wales, Australia.*
[3]*Marine Ecology Research Centre, Southern Cross University, P.O. Box 157, Lismore, NSW 2480,*
*Australia University, P.O. Box 157, Lismore, NSW 2480, Australia.*
[4]*Laboratório de Biogeoquímica, Universidade Federal do Rio de Janeiro (UFRJ), Rio d  Janeiro (RJ),*
*21941 971, Brazil.*
[5]*Department of Environmental Change, Linköping University, 581 83, Linköping, Sweden.*
[6]*Ecosystems and Global Change Laboratory (LEMG-UFF) / International Laboratory of Global Change*
*(LINCGlobal). Biomass and Water Management Research Center (NAB-UFF). Graduated*
*Program in Geosciences (Environmental Geochemistry). Universidade Federal Fluminense*
*(UFF), Av. Edmundo March, s/nº – Zip Code: 24210-310, Niteroi/RJ- Brazil.*
[7]*Sedimentary and Environmental Processes Laboratory (LAPSA-UFF). Department of Geography.*
*Graduated Program in Geography. Universidade Federal Fluminense (UFF), Av. Gal. Milton*
*Tavares de Souza, s/nº - Zip Code: 24210-346, Niteroi/RJ- Brazil.*
*Corresponding author. E-mail address; l.sanders.13@student.scu.edu.au






**Abstract**
The forests along the Amazon Basin produce significant quantities of organic
material, a portion of which is deposited in floodplain lakes. However, potentially
important effects of ongoing deforestation in the watershed on these carbon fluxes is still
poorly understood.   Here, a sediment core was extracted from an Amazon floodplain
lake to examine the relationship between carbon burial and land cover/use. Historical
records from 1942 and satellite data from 1975 were used to calculate deforestation rates
between 1942 and 1975, and 1975 to 2008 in four zones with different distances from the
margins of the lake and its tributaries (100, 500, 1000 and 6000-m buffers). Sediment
accumulation rates were determined from the $^{240+239}$Pu signatures and the excess $^{210}$Pb
method, reaching near 3.8 and 4.2 mm year$^{-1}$ in the last 60 and 120 years respectively.
The average carbon burial rates ranged between 100 and 350 g C m$^{-2}$ year$^{-1}$, with pulses
of high carbon burial derived from the forest vegetation, as indicated by $\delta^{13}$C and $\delta^{15}$N
signatures, which corresponded to heavy deforestation in the 1940 and 50s. Finally, our
results revealed a potentially important spatial dependence of the OC burial in Amazon
lacustrine sediments in relation to deforestation rates in the catchment. These
deforestation rates were more intense in the riparian vegetation (100-m buffer) during the
period 1942-1975 and the larger open water areas (500, 1000 and 6000-m buffer) during
1975-2008. The continued removal of vegetation from the interior of the forest was not
related to the peak of OC burial in the lake, but only the riparian deforestation around
1950. Our novel findings suggest the importance of abrupt and temporary events in which
some of the biomass released by the deforestation, especially restricted to areas along
open water edges, might reach the depositional environments in the floodplain of the
Amazon Basin.





## 1. Introduction

Rivers act as vectors, transporting sediment from land to ocean (Abril et al. 2014). Along this trajectory a significant proportion of the sediment load, including organic material, may be deposited in floodplains, creating zones of carbon accumulation (Smith et al. 2002, Dong et al. 2012, Hoffmann et al. 2013). This process is accelerated during flood events, when rivers and tributaries deposit organic material along the inundated floodplains (Smith et al. 2002). In some climate zones floodplains are seasonally inundated, with riparian zone vegetation dependent upon this seasonal influx of organic material. The vegetation acts to slow water velocity and trap the fine-grained, carbon rich sediment, within the low-energy environment (Aalto et al. 2003). Therefore, the riparian vegetation along her floodplains may be important for the organic matter deposition and the Amazon carbon cycle.

The importance of tropical wetland ecosystems in the carbon cycle is well documented (Downing et al. 1993, Melack et al. 2004, Zocatelli et al. 2013, Abril et al. 2014, Marotta et al. 2014). It has been shown that wetlands in the warm tropics are some of the most productive biological communities in the world (Neue et al. 1997), representing an important sink for nutrients (Marotta et al. 2009) and carbon (Peixoto et al. 2016), as well as sources of organic substrates to carbon gas production in inland waters (Marotta et al. 2010). However, these wetland ecosystems are also highly threatened by land use activities, especially from deforestation, development of agricultural land and soil degradation (Junk 2013, Lucas et al. 2014). For example, the Amazon Basin wetlands are being degraded by farming activities such as commercial ranching, and an increase in road density (Goulding 1993).

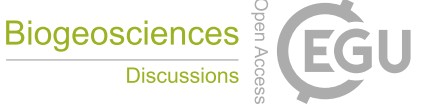

Deforestation of the Amazon Basin accelerated toward the end of the 1970's
(Skole and Tucker 1993), when an estimated 15% of the pristine rainforest area was lost
by the year 2003, increasing to approximately 18% by 2015 (INPE 2016). The ongoing
loss of vegetation is responsible for a substantial increase in erosion rates and subsequent
sediment inputs into Amazon rivers and lakes (Neill et al. 2013b). Yet these
anthropogenic activities are potential sources of allochthonous organic matter that may
increase carbon stores in the associated floodplain areas (Diaz and Rosenberg 2008,
Stanley et al. 2012).
The city of Santarem, in central Amazon, was established in the mid-eighteenth
century, approximately 650 km upstream from the Amazon River mouth and at its
confluence with the River Tapajós (02°25'0.28"S and 54°42'41.57"W, Figure 1). In 1940,
Santarém was only a small village with less than 0.5 km$^2$, surrounded by dense pristine
rainforest (estimation from the historical mapping of the Santarém City Hall). This city
quickly expanded, occupying 5.2 km$^2$ by the end of the 1970s and 49.3 km$^2$ currently
(estimation from satellite images LANDSAT/SRTM). Jupindá Lake is 70 km East of
from Santarém City, and receives surface water inflow from small streams draining from
the forest and the main tributary Curuá-Una River, a large affluent of the Amazon River
(Figure 1). The Lake has been affected by the deforestation associated with the expansion
of Santarém City. Between the 1940's and 1950's, there was intense deforestation on the
margins of rivers and streams in this area, used to supply the market of wood and forestry
products (Amorim 2000, Cruz et al. 2011). In the 1970s, the Curuá-Una River was
dammed (Curuá-Una Dam) 45 km upstream of Jupindá Lake to build the first
hydroelectric plant of the Amazon Forest (LigockI 2003).



Jupindá Lake provides an ideal opportunity to investigate historical changes in
organic carbon burial in a floodplain lake as a result of anthropogenic activities. This will
aid identification of still-little known impacts of land cover changes on recent carbon
burial rates in depositional environments of the Amazon floodplain.
The objectives of this research are to investigate the affect deforestation and urban
development has on carbon burial rates in a tropical floodplain wetland.


**2. Methods**
A 60 cm depth sediment core (diameter 7.5 cm) was collected in 2010 using a
gravity corer in the center of the Jupindá Lake (02°27'43.60" S, 54° 5'1.30" W. The
sediment core was sub sampled at 2 cm intervals. Dry bulk density (DBD, g cm$^{-3}$) was
determined as the dry sediment weight (g) divided by the initial volume (cm$^3$).    A
homogenized portion was acidified to remove carbonate material, then dried and ground
to powder for organic carbon (OC), nitrogen (N), $\delta^{13}$C, and $\delta^{15}$N analyses using a Flash
Elemental Analyzer coupled to a Thermo Fisher Delta V IRMS (isotope ratio mass
spectrometer). Analytical precision: C = 0.1 %, N = 0.1%, $\delta^{13}$C = 0.1‰ and $\delta^{15}$N = 0.15
‰.
Samples were prepared for Pu dating following the method of Ketterer et al.
(2004) with modifications to enable larger sample mass to be processed as  a result of the
likely lower Pu concentrations in the Southern Hemisphere (Sanders et al. 2014). To
obtain a larger mass, sediment intervals were joined and homogenized so the sediment
intervals for the $^{240+239}$Pu dating was 4 cm intervals.  Sample aliquots ranging from 14 to



grams were dry-ashed at 600 $^{o}$C for 16 hours, and leached with 50 mL of 16 M $HNO_3$.
The leaching was conducted overnight at 80$^{o}$C with added $^{242}$Pu yield tracer (NIST
4334g, 19 picograms). Acid leaching (as opposed to complete dissolution with HF) is
known to solubilize stratospheric fallout Pu, and there is little possibility that "refractory"
$HNO_3$-insoluble Pu exists in the South America (Sanders et al. 2014). The leachates
were diluted to 100 mL, filtered to remove solids, and the aqueous solutions were
processed with TEVA resin (EIChrom, Lisle, Il, USA) in order to chemically isolate 3.0
mL Pu fractions in aqueous ammonium oxalate solution suitable for measurements by
sector ICPMS. Pu determinations were performed using a VG Axiom MC operating in
the single collector (electron multiplier) mode. The system was used with an APEX HF
desolvating micronebulizer system (ESI Scientific, Omaha, NE, USA) with an uptake
rate of 0.4 mL/minute. Qualitative mass spectral scans (averages of 50 sweeps over the
mass range 237.4 – 242.6) were collected for selected samples prior to the electrostatic
sector quantitative scanning of $^{238}$U+, $^{239}$Pu+, $^{240}$Pu+, and $^{242}$Pu+. Detection limits were
evaluated based upon the analysis of two blanks and considerations regarding the
obtained mass spectra. A detection limit of 0.01 Bq/kg of $^{239+240}$Pu is applicable for
samples of nominal 25 g mass.

142        For $^{210}$Pb dating, an intrinsic germanium detector coupled to a multi-channel

analyzer was used. Freeze dried and ground sediments were packed and sealed in gamma
tubes. Lead-210 and $^{226}$Ra activities were calculated by multiplying the counts per minute
by a factor that includes the gamma-ray intensity and detector efficiency determined from
standard calibrations. Identical geometry was used for all samples. Lead-210 activities
were determined by the direct measurement of the 46.5 KeV gamma peak. Radium-226





activity was determined via the $^{214}$Pb daughter at 351.9 KeV. For $^{226}$Ra measurements,
the packed samples were set aside for at least 21 days to allow for $^{222}$Rn to ingrow and
establish secular equilibrium between $^{226}$Ra and its granddaughter $^{214}$Pb. Excess $^{210}$Pb
activity was calculated by subtracting the supported $^{210}$Pb (i.e., $^{226}$Ra activity) from the
total $^{210}$Pb activity. The sediment accretion rate for the previous 120 years was estimated
by two methods derived from $^{210}$Pb dating, the Constant Initial Concentration (CIC)
model assuming that this rate has not varied during the encompassed time span (Appleby
and Oldfield 1992), and the Constant Rate of Supply (CRS) model based on a constant
influx of unsupported, atmospheric $^{210}$Pb that allows a variable sediment rate (Ivanovich
and Harmon 1992). Organic carbon accumulation rates were estimated from sediment
accretion rates (mm yr$^{-1}$), dry bulk density (g cm$^{-3}$) and OC content.

The land/use cover analysis was based on documented historical information

before 1975 and satellite images (Landsat/SRTM, Table 1) from 1975, 1985, 1995 and
2008 available from the United States Geological Survey (USGS). No significant
deforestation occurred in the catchment area of the Jupindá Lake  until early 1940's
(Amorim 2000, Cruz et al. 2011). Subsequent land/use changes were determined using
satellite images (Gordon 1980, Munyati 2000). All satellite images were from low-water
seasons to remove the influence of the flood pulse on the exposed area over years. The
resolution of the images was 30 m, except that from the 1970's which was resampled
from 90 to 30 m (Table 1). This approach allowed an assessment of changes in land cover
which could then be compared to results from carbon accumulation.  Results of the spatial
assessment were separated into two time periods; 1942-1975, or the timeframe between the
onset of land clearing and the first satellite image, and 1975-2008 which provides a more
detailed assessment of temporal changes to the study area. The time period 1942-1975 was



characterized by a rapid removal (peak until the 1960's) of vegetation established at the
margins of inland waters; especially *Aniba rosaeodora* (Pau-rosa) for extraction of oils,
and *Mezilaurus itauba* and *Cedrela fissilis* (Louro-itaúba and Cedro, respectively) as
hardwoods, and the opening of clearings for crops of textile fibers and subsistence
products. Further, intensification of deforestation towards the interior of the forest and
following the urban growth of Santarém is reported from the 1970's, along with depleting
vegetal resources near to the margins of lakes and running waters in this region was noted
(Amorim 2000, Cruz et al. 2011).

In order to address the spatial dependence of recent OC burial in Jupindá Lake for

deforestation, we analyzed the land/cover use in four buffer areas around this lake and
contributing rivers or streams. The first buffer of 100 m represented the riparian forest
protected area by the Brazilian laws for fluvial channels with a width of 50 to 200 m.
Other buffers were progressively higher, with a width of 500, 1000 and 6000 m from the
riverbank and lake margins (Figure 2). In addition, we considered only stretches of rivers
and streams 65-km long from Jupindá Lake to analyze its catchment area of more direct
influence. This criteria also allowed to avoid the interference of the artificial flooding on
the margins of the Curuá-Una hydroelectric dam, which was built in 1977 (Fearnside

2005).



**3. Results**

Analyses of $^{239+240}$Pu were not detectable from the bottom of the sediment core until

the 22-26 cm interval (Figure 43). This radioisotope was detected in the 18-22 cm



interval (0.029 ± 0.002 Bq/kg $^{239+240}$Pu) with the highest concentrations (0.047 ± 0.004
Bq/kg $^{239+240}$Pu) at the 16 cm depth. The $^{239+240}$Pu activities appears to spike at the 14 to
18 cm interval, which indicates the 1963 stratospheric fallout peak. It may be said with
certainty that the material below 22 cm was deposited pre-bomb (that is, prior to the early
1950's). This affixes an upper limit on the average sedimentation rate of near to 3.8 mm
year$^{-1}$. The Pu atom ratio data indicate that the Pu is originating from stratospheric
fallout (plutonium isotopic ratios ($^{240/239}$Pu) of ~0.18). These results are consistent with
the $^{240}$Pu$^{/239}$Pu of 0.180 ± 0.014 discussed by Kelley et al. (1999).

The $^{210}$Pb and $^{226}$Ra profiles reveal a complex depositional environment with

sedimentation variations in the upper intervals with disturbances, such as bio-turbation
and resuspension in the upper ~ 20 cm of the sediment column (Figure 4). A decrease in
$^{210}$Pb$_{ex}$ activity was found below the 20 cm depth interval. The $^{210}$Pb$_{ex}$ data distribution
are as follows: y = -0.0749x + 7.5) ($R^2$ = 0.73; n=19; p < 0.01) from the 20 to the 60 cm
interval, below the apparent mixed zone. Both estimates of sediment accretion rate during
the 120 years from CIC and CRS models were similar, reaching 4.1 and 4.3 mm yr$^{-1}$
respectively, which were   slightly higher than the ~ 60 year $^{239+240}$Pu dates (3.8 mm yr$^{-1}$).

The dry bulk density (DBD), total organic carbon (OC%), total nitrogen (TN%)

content and the carbon and nitrogen (C/N) molar ratios along with the $\delta^{13}$C  and  $\delta^{15}$N
values showed important increases towards the center of the sediment core Table 2. The
relationship between $\delta^{13}$C and $\delta^{15}$N indicated different origins of OC in the sediment core
(Figure 5) contributing to the significant relationship between recent OC burial and the
$\delta^{13}$C (Figure 6).

The OC burial rates show an increasing trend from ~ 1930 to 1960 with a peak during



the 1940's and 50's (grey area in Figure 7).  The carbon burial rates increased, from 150
g m$^{-2}$ year$^{-1}$ in the time period 1890 - 1940 to ~ 300 g m$^{-2}$ year$^{-1}$ between 1940 and 1950.
Carbon accumulation then decreased to approximately 200 g m$^{-2}$ year$^{-1}$ from 1960 to
1980, after which a gradual decline in carbon burial was still measured. In relation to land
use/cover in the surroundings of fluvial channels and the Jupindá lake over time, only the
smallest buffer (100 m) showed more intense relative changes during the previous period
1942-1975, when the increase in deforested area was around 75 % higher than in the
subsequent time period 1975-2008 (Figure 8).

**4. Discussion**

Overall, similar estimates of sediment accretion using different methodologies

(i.e. 60 and 120 year trends from the $^{239+240}$Pu and $^{210}$Pb$_{(ex)}$ models, respectively) revealed
an insight into changes in the sediment accumulation assessed here.    This indicates that
even though the origin of the organic material was modified, the sediment accumulation
has varied little as indicated by the 60- or 120-year sediment accumulation rates.  To
present the historical profiles of the carbon burial rates, an average is taken between these
two methods (4 mm year$^{-1}$), multiplied by the DBD and carbon content for each interval
of the entire sediment core.

The high peak in carbon accumulation observed around 1950 appears to be

associated with a shift in the source of organic material, inferred by changes in carbon
and nitrogen contents and the isotopic fractioning toward the middle (from 40 to 20 cm
depth interval) of the sediment column. This peak for different organic and inorganic
variables in intermediate depths revealed changes not only in the amount but also in the





type of material being deposited over time. Previous studies have reported two common
origins for OC in the Amazon forest. Higher $\delta^{15}$N and more negative $\delta^{13}$C values could
indicate the presence of Santarém soil organic matter (such as that adjacent to the Jupindá
Lake), while lower $\delta^{15}$N and more variable $\delta^{13}$C values indicate particulate organic
carbon (POC) from the terrestrial vegetation in the catchment (Ometto et al. 2006,
Zocatelli et al. 2013). Here, a corresponding  increase in OC%, TN% and OC burial rates
measured, with a peak near ~1950,  suggesting higher inputs of organic matter into lake.
The higher $\delta^{13}$C signature, coupled with a lower $\delta^{15}$N indicates a greater influence from
the terrestrial Amazonian POC during the same period around 1950 (Ometto et al., 2006).

When looking for a cause for this change in the source of organic material, we

look to the analysis of land use change.  Land clearing associated with early occupation
from the 1940s in the catchment area of the Jupindá Lake reveals a potential cause of the
increased carbon burial observed in this lake. Changes in land use and cover may
significantly affect recent OC burial in mid-high-latitude lakes (Anderson et al. 2013,
Dietz et al. 2015). Our results suggest that land clearing during the 1940's and 50's might
be related to increased organic matter deposition in the region's floodplain lakes.  During
this period, intense wood extraction and expansion of agricultural settlements occurred
(Amorim 2000, Cruz et al. 2011). One important consequence of deforestation in the
watershed is the silting up of lakes (Enea et al. 2012 ), including those at humid low-
latitude areas (Cohen et al. 2005, Bakoariniaina et al. 2006).  The riparian forest systems
are generally effective in reducing the sediment transport by surface runoff, with the
removal of this vegetation increasing the erosion processes especially in the Amazon
basin due to intense rainfall (Neill et al. 2013a).



We also found a spatial dependence of the carbon accumulation in the Lake
Jupindá, as the much lower OC burial was coupled to higher deforestation rates in those
larger buffers around its margins and main fluvial channels (500, 1000 and 6000 m) in
the period after 1975 (1975-2008) than that before (1942-1975). This confirms previous
evidences that the recent deforestation process in the region was started in areas around
running and lake waters (Amorim 2000, Cruz et al. 2011), and not in the interior of the
forest. The enhanced OC burial in lacustrine sediments before 1975 was related to higher
deforestation rates only in the riparian vegetation zone (100-m buffers), suggesting a
higher influence of deforestation with decreasing distance to water courses. Therefore,
the soil carbon enrichment to the aquatic sediments during the peaks of riparian
deforestation may cause intense but temporary carbon burial events in the Amazon
floodplain, representing a significant part of the total loss of terrestrial organic matter. In
contrast, the continued removal of vegetation from the interior of the forest might be not
directly related to increases of OC burial, even temporarily, in depositional aquatic
ecosystems.

**5. Conclusion**
The $^{239+240}$Pu and $^{210}$Pb dating methods were combined with a spatial analysis of
vegetation clearing to firstly calculate carbon accumulation rates, and then to interpret
changes in sediment characteristics during the previous century.  The Pu dating method
closely approximates measurements from the $^{210}$Pb chronologies and hence offers
mechanism to determine sedimentation rates and carbon accumulation in Amazon
sediments. An increase in OC burial, 150 to ~ 300 OC g m$^{-2}$ year$^{-1}$, coincides with



changes in the $\delta^{13}$C and $\delta^{15}$N signatures, likely influenced by the heavy deforestation in
riparian systems of this region during the 1940s and 50's. It is therefore suggested that
the net increase in carbon burial towards the center of the sediment core, which
represents the highest carbon burial rates during the 1950s, is a result of a change in
source of organic matter deposition. The differing carbon burial rates along the sediment
core reveals the potential complexity of carbon burial rates in the Amazon floodplain
lakes, directly related to the development within the Basin. This work demonstrates a
new understanding on spatial dependence of carbon burial capacity of the Amazon
floodplain lakes with respect to advances in deforestation in the basin.


**Acknowledgements**
LMS is supported by an APA and IPRS scholarships. HM received a research grant from
the Brazilian Research Council (CNPq – "Programa Universal") and the Research
Support Foundation of the State of Rio de Janeiro (FAPERJ – "Programa Jovem Cientista
do Nosso Estado").


**CAPTIONS TO FIGURES**
**Figure 1.** Floodplain Lake where the sediment core was collect, near the Amazon River
and the city of Santarém, Brazil. This floodplain lake has a diameter of approximately 3
km.





**Figure 2.** Different buffer sizes (100m, 500m, 1km and 6km) along the stretch of the
Curuá-Una river from Jupindá Lake (red) to the hydroelectric dam upstream (yellow).
**Figure 3.** $^{239+240}$Pu profile, indicating ~ 1950 when these radionuclides were first
introduced into the atmosphere.
**Figure 4.** Excess $^{210}$Pb and $^{226}$Ra profile against depth.
**Figure 5.** $\delta^{13}$C vs $\delta^{15}$N. The Amazon River POM and Santarem soil organic matter
values, adjacent to the study area, are taken from Zocatelli et al (2013).
**Figure 6.** Carbon burial as a function of $\delta^{13}$C.
**Figure 7**. $\delta^{13}$C, $\delta^{15}$N and carbon burial rate values in relation to age (year).
**Figure 8.** Percentage of modified areas in relation to the different buffers.


**CAPTION TO TABLES**
**Table 1**. Satellite acquisition data from United States Geological Survey (USGS) and the
Curuá-Una River quota from Brazilian Water Agency (ANA).
**Table 2.** Depth profiles of dry bulk density (DBD), total organic carbon (OC%), total
nitrogen (TN%) carbon and nitrogen (C/N) molar ratios, $\delta^{13}$C and $\delta^{15}$N.
**Table 1.**

| *Month/Year* | *Landsat Data* | *Curuá-Una River Quote* |
|---|---|---|
| Aug/1975 | 2 | 5.3 |
| Oct/1985 | 5 | 3.7 |
| June/1995 | 5 | 6 |
| June/2008 | 5 | *No data* |




**Table 2.**

| Depth (cm) | DBD (g cm$^{-3}$) | $\delta^{15}$N | $\delta^{13}$C | C (%) | N (%) | C/N |
|---|---|---|---|---|---|---|
| 0-2 | 1.0 | 8.9 | -29.2 | 3.8 | 0.3 | 17.2 |
| 2-4 | 0.9 | 11.7 | -29.0 | 3.8 | 0.3 | 18.7 |
| 4-6 | 1.0 | 10.4 | -28.8 | 4.0 | 0.3 | 19.2 |
| 6-8 | 1.1 | 9.3 | -28.7 | 4.3 | 0.3 | 20.2 |
| 8-10 | 1.0 | 9.4 | -28.7 | 4.1 | 0.3 | 19.8 |
| 10-12 | 1.1 | 7.9 | -28.6 | 4.6 | 0.3 | 21.2 |
| 12-14 | 1.1 | 8.2 | -28.7 | 4.3 | 0.3 | 19.9 |
| 14-16 | 1.1 | 7.8 | -28.6 | 4.3 | 0.3 | 20.9 |
| 16-18 | 1.0 | 8.7 | -28.5 | 4.4 | 0.3 | 21.2 |
| 18-20 | 1.1 | 7.5 | -28.4 | 4.4 | 0.3 | 19.8 |
| 20-22 | 1.0 | 6.5 | -28.2 | 5.4 | 0.3 | 21.2 |
| 22-24 | 1.0 | 6.0 | -27.8 | 5.3 | 0.3 | 21.5 |
| 24-26 | 1.0 | 5.2 | -27.4 | 7.3 | 0.4 | 25.4 |
| 26-28 | 1.1 | 6.1 | -27.6 | 6.0 | 0.3 | 23.8 |
| 28-30 | 1.0 | 5.0 | -27.3 | 6.0 | 0.4 | 22.7 |
| 30-32 | 1.0 | 5.4 | -28.0 | 6.1 | 0.3 | 27.0 |
| 32-34 | 1.3 | 6.6 | -28.5 | 4.4 | 0.2 | 27.5 |
| 34-36 | 1.6 | 8.9 | -29.0 | 2.2 | 0.1 | 23.1 |
| 36-38 | 1.4 | 11.4 | -29.4 | 2.9 | 0.1 | 30.4 |
| 38-40 | 1.4 | 10.4 | -29.5 | 3.3 | 0.1 | 30.5 |
| 40-42 | 1.5 | 11.4 | -29.3 | 2.4 | 0.1 | 23.8 |
| 42-44 | 1.6 | 12.2 | -29.4 | 1.3 | 0.1 | 15.6 |
| 44-46 | 1.8 | 8.2 | -29.6 | 1.2 | 0.1 | 14.3 |
| 46-48 | 1.5 | 8.8 | -29.8 | 2.2 | 0.1 | 21.6 |
| 48-50 | 0.9 | 10.4 | -29.7 | 2.9 | 0.2 | 25.6 |
| 50-52 | 0.9 | 10.2 | -29.7 | 2.6 | 0.1 | 27.2 |
| 52-54 | 0.9 | 7.1 | -29.7 | 3.9 | 0.2 | 28.6 |
| 54-56 | 0.9 | 9.2 | -29.9 | 3.6 | 0.2 | 27.8 |
| 56-58 | 0.9 | 6.6 | -30.1 | 4.3 | 0.2 | 30.1 |
| 58-60 | 0.9 | 5.0 | -30.1 | 3.5 | 0.2 | 23.1 |
| **Average** | **1.11** | **8.34** | **-28.9** | **4.0** | **0.2** | **23.0** |
| **Stand Dev** | **0.24** | **2.1** | **0.8** | **1.9** | **0.1** | **4.2** |




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



**Figure 1.**

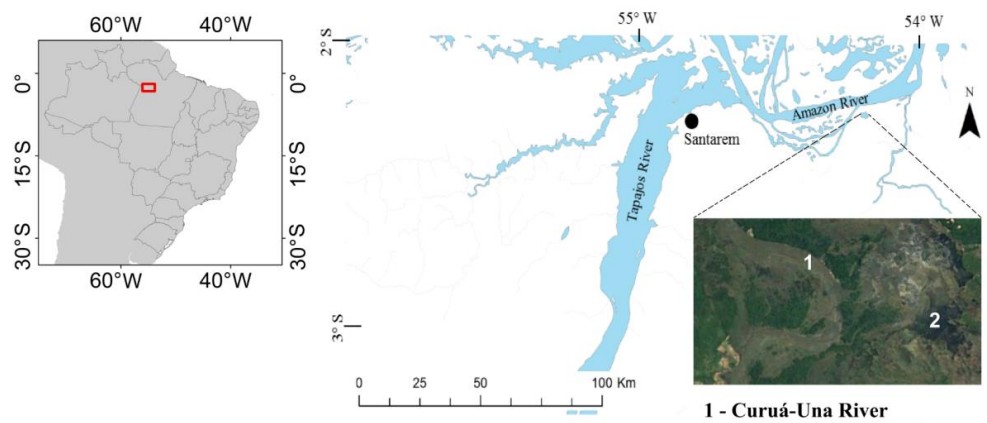




**Figure 2.**

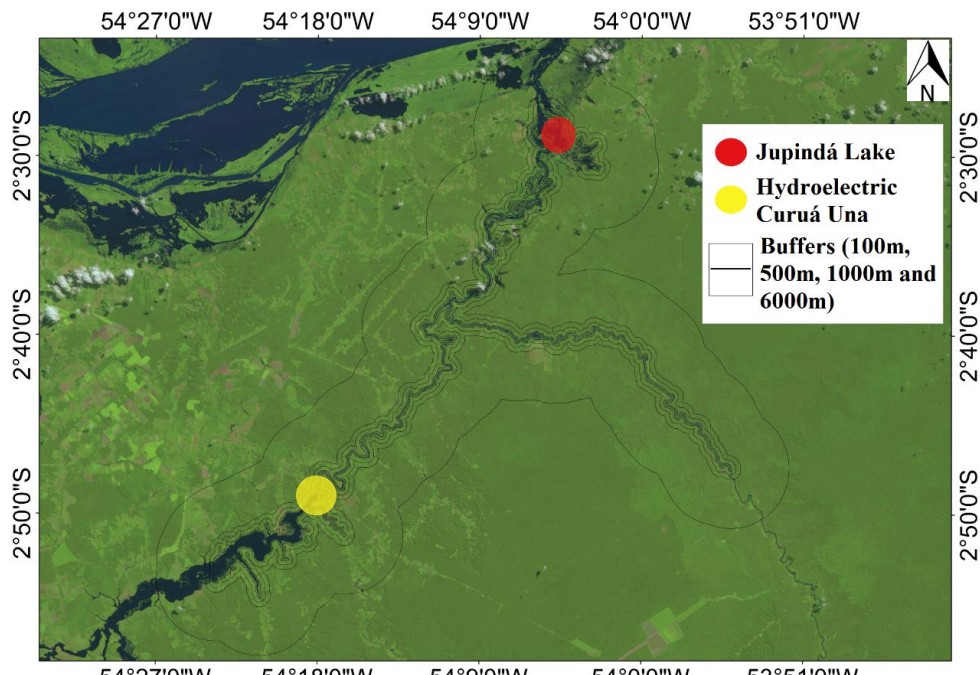


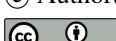

**Figure 3.**

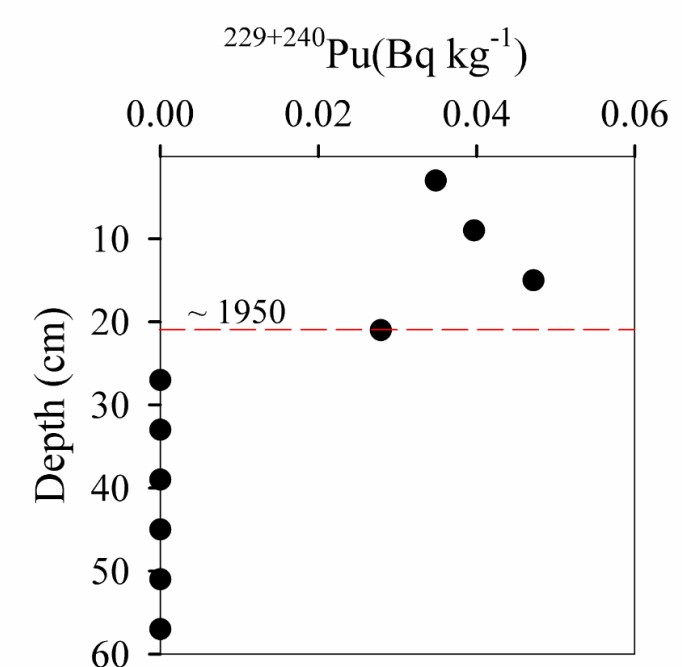






**Figure 4.**

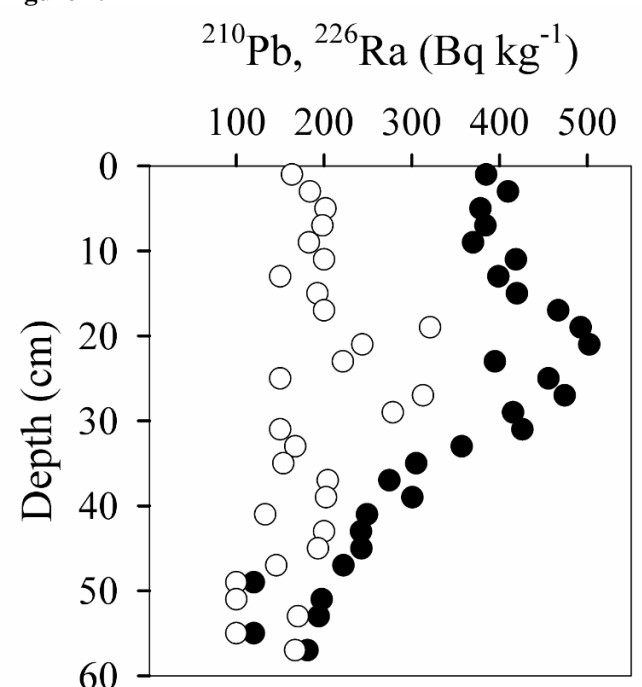






**Figure 5.**

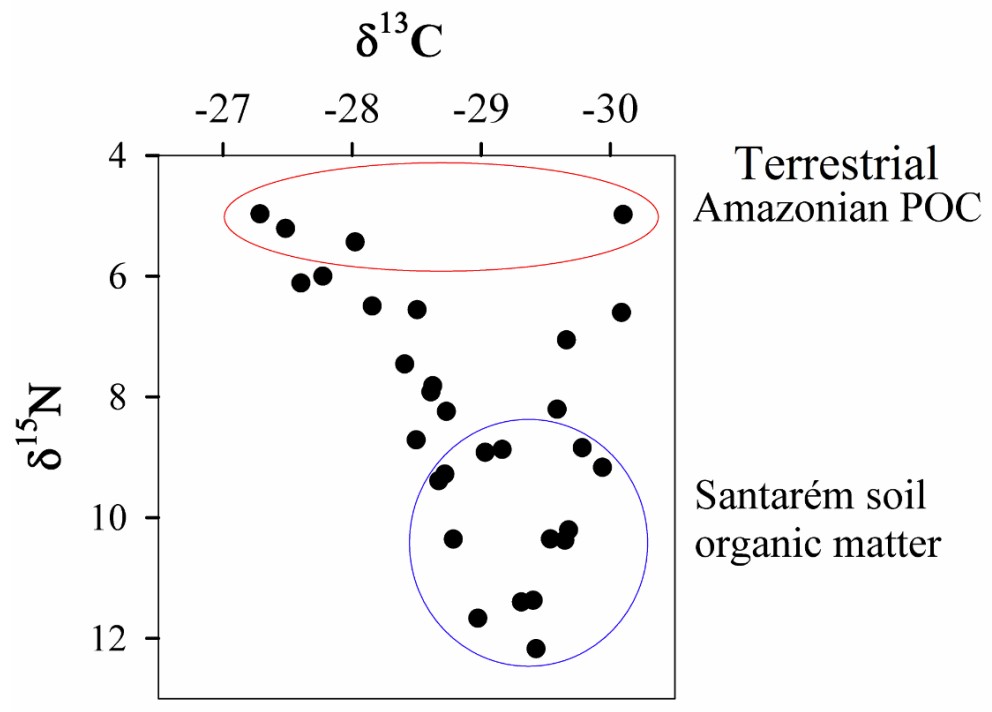




**Figure 6.**

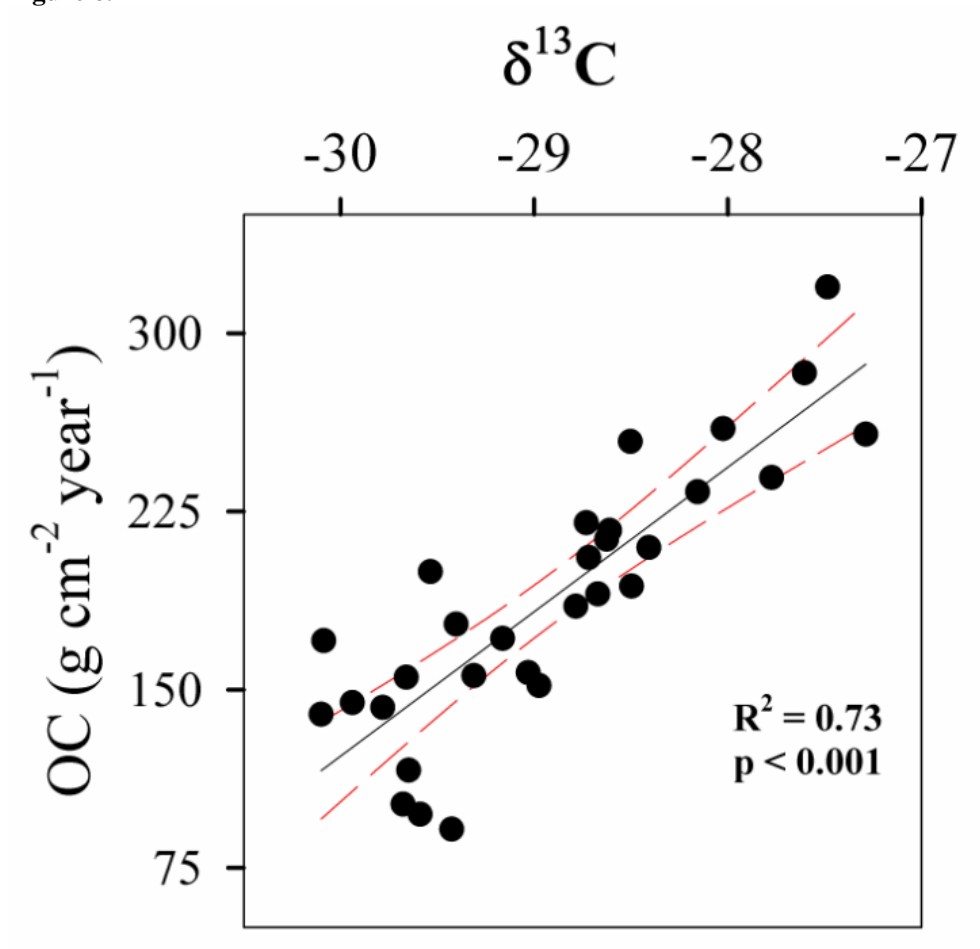




**Figure 7.**

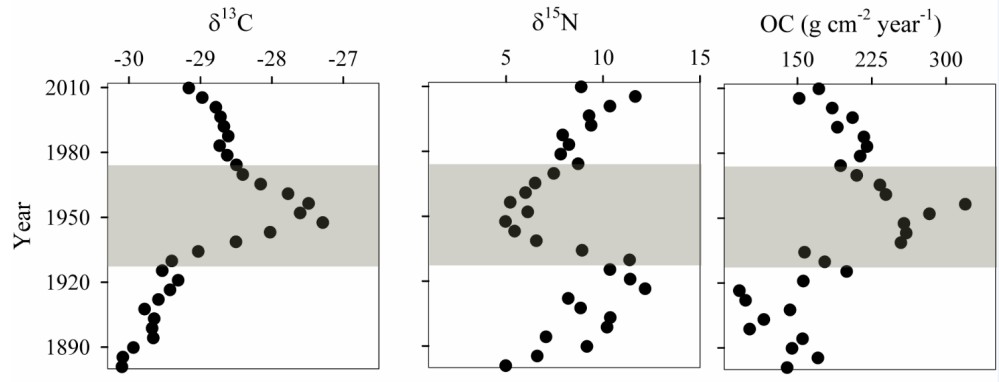




**Figure 8.**

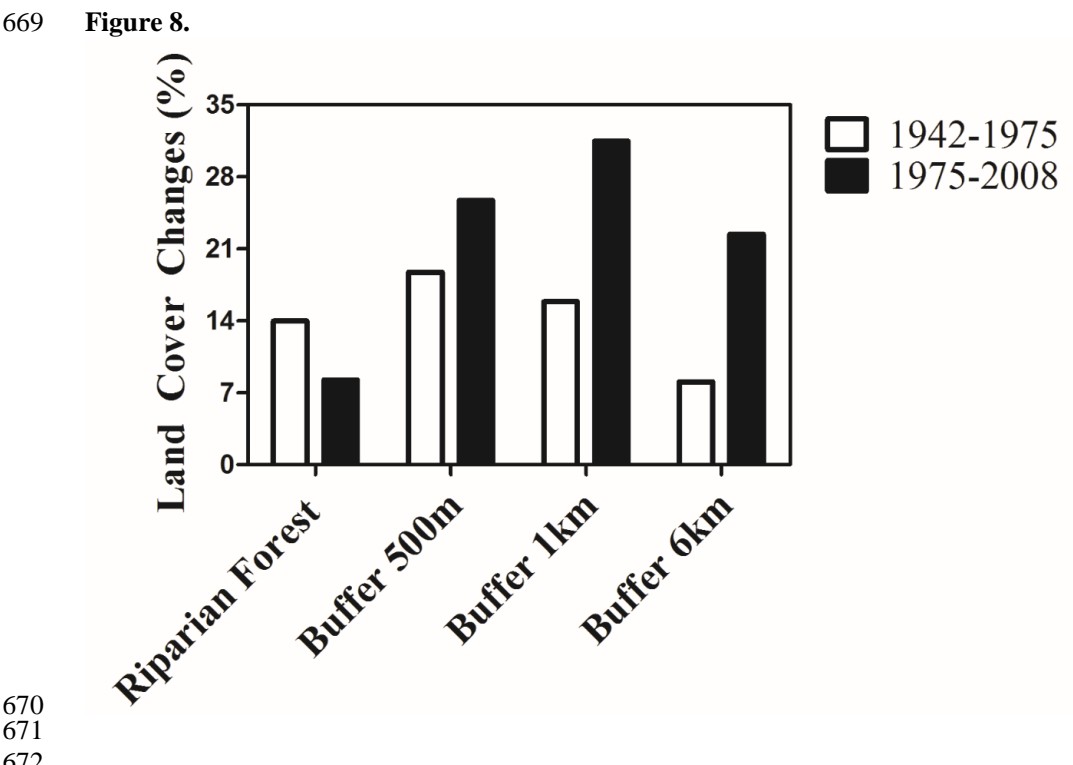
