# Peer review of "Historic carbon burial spike in an Amazon floodplain lake linked to riparian deforestation near Santarem, Brazil"

_Biogeosciences, 2017_

## Referee Comment (RC1) · JA Hutchings (Referee) · 21 Jul 2017

General Comments: Sanders and co-authors analyzed a core from an Amazon floodplain lake in order to assess whether the lake's C accumulation is related to anthropogenic changes in the region. This manuscript does a good job of introducing the topic and appears to have performed data collection in an appropriate manner. However, the ultimately analyses of collected data are lacking, and I recommend the authors to further their exploration of this data set. In particular, additional or alternative plots showing mass accumulation rates as well as cumulative burial would be welcome. The authors attribute changes in carbon burial rates to anthropogenic disturbance, but

neither back this claim up in a statistically rigorous manner nor do they seriously (i.e., in a statistically rigorous manner) consider alternative causes for their burial rate changes. These problems of data analysis need to be resolved before the merits of this study can be fully judged. Specific Comments: Line 213: I am not sure what is meant by 'important' here, consider clarifying or removing. Line 214-216: Given that a much better separation is found between d15N and OC source (Figure 6), why not regress d15N against the instantaneous C accumulation rate as you did for d13C in Figure 7? Line 232-235: This bit about data processing should likely be in the methods. Line 259: Do you have evidence of the 'silting up' of this lake? Perhaps in the form of mass accumulation rates rather than OC, or, alternatively, looking for changes in DBD with depth. Line 264-272: An ANOVA (or similar approach) would be an appropriate test here to help put some weight behind this statement. Figures 6 and 7: The C accumulation rates should probably be g/m2/yr, right?

---

## Author Comment (AC1) · 4 Aug 2017

JA Hutchings (Referee) jahutch2@gmail.com

Please see attached file (Supplement PDF) for revised manuscript.

General Comments: Sanders and co-authors analyzed a core from an Amazon floodplain lake in order to assess whether the lake's C accumulation is related to anthropogenic changes in the region. This manuscript does a good job of introduc-

ing the topic and appears to have performed data collection in an appropriate manner. However, the ultimately analyses of collected data are lacking, and I recommend the authors to further their exploration of this data set.

We appreciate the positive feedback. Below we respond to each comment individually.

In particular, additional or alternative plots showing mass accumulation rates as well as cumulative burial would be welcome.

Response: We agree and have included an excess 210Pb(ex) profile vs cumulative dry mass figure (see Figure 4). We also include additional plots showing the statistical analyses of the major finding found in this work (see Figure 7).

The authors attribute changes in carbon burial rates to anthropogenic disturbance, but neither back this claim up in a statistically rigorous manner nor do they seriously (i.e., in a statistically rigorous manner) consider alternative causes for their burial rate changes. These problems of data analysis need to be resolved before the merits of this study can be fully judged.

Response: We agree with the Reviewer, and we have included additional statistical approaches to test the differences between the peak in OC burial with the periods before and after the deforestation period, by adding new panels to Figure 7, showing the statistical differences among these phases. The Figure 7 captions now contain the following: "Panels below each vertical profile represents respective data grouped by the phases >1934, 1934-1975 and 1975-2008. Filled square symbols represent means of a given variable in each sediment layer, and the vertical bars show the mean with the standard deviation of the respective phase. Equal letters in each panel represent non-significant differences ($p > 0.05$, one-way ANOVA followed by Tukey's post test)."

Furthermore we have added the following the Discussion section (Line 258): "The stable isotope results and OC burial rates, when grouped into different phases, showed assumptions required for parametric analyses, including normal distribution (Kolmogorov-

Smirnov, p > 0.05) and homogeneity of variance (Bartlett, p > 0.05) (Figure 7). Thus, when examining the means and standard errors to represent the distribution of values, and parametric tests, different sedimentary phases are noted. These different sedimentary phases are confirmed by statistical differences as tested using a one-way ANOVA test followed by Tukey's post test (significance was defined as p < 0.05). " Also, we highlighted that this lake is situated in a region relatively preserved in the Amazon Forest, and that another explanation than deforestation in the margins to the peak of sedimentation might be not reasonable. To highlight these this, we now include the following (Line 279): " The lake is in a region relatively preserved, and therefore there is no other explanation other than deforestation in the margins which may have caused the peak in OC burial found between 1934 and 1975."

Specific Comments: Line 213: I am not sure what is meant by 'important' here, consider clarifying or removing.

Response: We agree and "important" was replaced by "an" to read (Line 213): "showed an increases towards the center of the sediment core".

Line 214-216: Given that a much better separation is found between d15N and OC source (Figure 6), why not regress d15N against the instantaneous C accumulation rate as you did for d13C in Figure 7?

Response: We agree and d15N against OC accumulation panel has been added to this Figure 6.

Line 232-235: This bit about data processing should likely be in the methods.

Response: We agree and have added the follow to Methods section (Line 156): "Organic carbon accumulation rates were estimated from an average between these the two dating methods, 239+240Pu and 210Pbex the dry bulk density (g cm-3) and carbon content for each interval of the entire sediment core."

Line 259: Do you have evidence of the 'silting up' of this lake? Perhaps in the form of mass accumulation rates rather than OC, or, alternatively, looking for changes in DBD with depth.

Response: Mass accumulation rates, directly related to changes in DBD with depth, indicate a change in sediment source. However, this lake is situated in a region relatively preserved in the Amazon Forest. We now include the following (Line 279): "The lake is a region relatively preserved, and therefore there is no other explanation other than deforestation in the margins which may have caused the peak in OC burial found between 1934 and 1975."

Line 264-272: An ANOVA (or similar approach) would be an appropriate test here to help put some weight behind this statement.

Response: We agree and tested statistical differences (one-way ANOVA followed by the Tukey's post test) among phases of sedimentation. Se new Fig 7 and captions. We also have added information on the statistical analyse throughout the manuscript, including on the sentence added in the Results section of (Line 220): " The peak of greater $\delta$13C and lower $\delta$15N values coupled to higher OC burial rates were observed in the phase between 1934-1975 in Jupindá Lake (one-way ANOVA followed by Tukey's post test, p<0,05; Fig. 7). The $\delta$13C values were around 3 and 5% greater in the phase 1934-1975 in relation to those previous and after respectively (one-way ANOVA followed by Tukey's post test, p<0,05). This peak between 1934-1975 also showed delta $\delta$15N values around 30% lower and OC burial rates around 40 % higher than other phases (one-way ANOVA followed by Tukey's post test, p<0,05)."

Figures 6 and 7: The C accumulation rates should probably be g/m2/yr, right? Response: The Reviewer is right and these corrections have been made to Figures 6 and 7 as suggested.

Once again, we thank the Reviewer for these constructive comments. We feel that these comments significantly strengthened our manuscript. END

Please also note the supplement to this comment:
https://www.biogeosciences-discuss.net/bg-2017-151/bg-2017-151-AC1-supplement.pdf

**Supplement:**

[revised manuscript text omitted]

---

## Referee Comment (RC2) · Anonymous Referee #2 · 8 Aug 2017

This study investigates changes in organic carbon burial in an Amazonian floodplain lake in response to deforestation and urban development. The authors conclude that OC burial was strongly affected by initial deforestation and development close to the river and lake margins, but not very strongly during later deforestation which took place further away from surface waters.

OC burial in lakes in an important but understudied carbon flux, and as such, studies like this are needed, particularly if coming from an understudied area of the world. Also, methodologically this study seems sound, and the conclusions seem plausible and supported.

[Figure]

I have, however, one major concern regarding the dating methods and the subsequent calculation of OC burial rates. The authors use two methods, measurement of plutonium (Pu) from atmospheric fallout after nuclear bomb testing, and measurement of 210Pb which is a naturally occurring atmospheric radioisotope. The Pu method returns a sediment depth corresponding to pre-bomb testing (∼1950), and therefore an average sediment accumulation rate since this time. The Pb method, on the other hand, can resolve changes in sediment accumulation rate over time, if combined with an appropriate model (e.g. the CRS model). Since OC burial is calculated as sediment accumulation rate * OC content * dry bulk density, the authors should use the Pb data. Instead, they calculate an average long-term mean sediment accumulation from the Pu and Pb data, and use this fixed, constant value for calculation of OC burial over time. It is not correct to present data on the temporal change of OC burial rates (Figs 6 and 7) that are based on keeping the most important term (sediment accumulation rate) constant.

I would like to encourage the authors to recalculate the OC burial values based on the Pb chronologies, and also to present these Pb chronologies in a graph. Alternatively, they should provide arguments as to why their approach is warranted.

In the following, I provide some more detailed comments.

87-101 This is the site description, and it should be moved to methods. Instead, elaborate here more about the ways you think deforestation and exploitation can affect OC burial in lakes, and mention and cite previous findings on this topic explicitly. Also, it would be good to see a clearly formulated, testable hypothesis spelled out in the Introduction.

L115. How was the acidification done? There are papers showing that different acidification methods affect the stable isotopic signature differently.

L 120. What was the motivation to use both Pu and Pb dating methods? Why not Pb alone? It seems that the Pu method is not suitable for resolving changes in sediment

accumulation rate over time, which is the purpose of this study. Is the sole purpose of the Pu method to derive an independent estimate of long-term average accumulation?

L194. Fig 43 must be a typo.

L204-206. Unclear what this means, and what the equation describes.

L217. Were the OC burial rates reported in the paragraph calculated using the CIC or CRS model on 210Pb data?

L221-225. This sentence is difficult to understand and long. Rephrase, please.

L228. I do not understand this reasoning. You have calculated 210Pb chronologies, and the activity profile (Fig.4) does show indications of changes in sediment accumulation rate over time. So why do you use an average long-term accumulation rate, based on the Pb method and the Pu method (which does not give any resolution in time of accumulation rate)? Why don't you trust the Pb chronologies? Using an average long-term accumulation rate is contrary to the purpose of your study. Here's also a conceptual flaw: if you present changes over time in OC burial rate (Figs 6 and 7), which do not account for changes over time in sediment accumulation rate, then these graphs only reflect changes in OC content and dry bulk density.

L264-267. It is difficult to see this linkage between the buffer zones, time periods and OC burial rates. Can you think of a way to illustrate these links in a graph?

L276-278. This is an important conclusion.

L283-284. But only concerning long-term average sediment accumulation rates, while the purpose of this study was to investigate temporal changes of sediment accumulation rate, which can only be accomplished with the Pb data. This distinction should be made throughout the ms, and the Pb chronologies using both the CIC and CRS models should be shown.

Figs 6 and 7. The axis label should be "OC burial rate", not just "OC".

---

## Author Response (AR1)

Note page and line numbers refer to the revised manuscript.

**Reviewer 1:** JA Hutchings (Referee) jahutch2@gmail.com

General Comments: Sanders and co-authors analyzed a core from an Amazon floodplain lake in order to assess whether the lake's C accumulation is related to anthropogenic changes in the region. This manuscript does a good job of introducing the topic and appears to have performed data collection in an appropriate manner. However, the ultimately analyses of collected data are lacking, and I recommend the authors to further their exploration of this data set.

We appreciate the positive feedback. Below we respond to each comment individually.

In particular, additional or alternative plots showing mass accumulation rates as well as cumulative burial would be welcome.

We agree and have included an excess $^{210}Pb_{(ex)}$ profile vs cumulative dry mass figure (see Figure 4). We also include additional plots showing the statistical analyses of the major finding found in this work (see Figure 7).

The authors attribute changes in carbon burial rates to anthropogenic disturbance, but neither back this claim up in a statistically rigorous manner nor do they seriously (i.e., in a statistically rigorous manner) consider alternative causes for their burial rate changes. These problems of data analysis need to be resolved before the merits of this study can be fully judged.

We agree with the Reviewer, and we have included additional statistical approaches to test the differences between the peak in OC burial with the periods before and after the deforestation period, by adding new panels to Figure 7, showing the statistical differences among these phases. The Figure 7 captions now contain the following: " Panels below each vertical profile represent respective data grouped by the phases >1934, 1934-1975 and 1975-2008.  Filled square symbols represent means of a given variable in each sediment layer, and the vertical bars show the mean with the standard deviation of the respective phase. Equal letters in each panel represent non-significant differences ($p > 0.05$, one-way ANOVA followed by Tukey's post test)."

Furthermore we have added the following the Discussion section (Line 272): "The statistical treatment of variables and OC burial rates, when grouped into different phases, showed assumptions which required parametric analyses, including normal distribution (Kolmogorov-Smirnov, $p > 0.05$) and homogeneity of variance (Bartlett, $p > 0.05$). Thus, we used means and standard errors to represent the distribution of values, and parametric tests were conducted to compare different phases. Statistical differences were tested using the one-way ANOVA test followed by Tukey's post test (significance was defined as $p < 0.05$)."

Also, we highlighted that this lake is situated in a region relatively preserved within the Amazon Forest, and other than deforestation in the margins there has been no disturbance in the area. To highlight this, we now include the following (Line 302): "The peak of the significantly greater $\delta^{13}C$ and lower $\delta^{15}N$ values coupled to higher OC burial rates were observed in the phase between 1934-1975 in Jupindá Lake (one-way ANOVA followed by Tukey's post test, p<0.05; Fig. 7). The $\delta^{13}C$ values were greater in the 1934-1975 phase as related to those previous and after respectively (one-way ANOVA followed by Tukey's post test, p<0.05). This peak between 1934-1975 also showed delta $\delta^{15}N$ values lower and OC burial rates higher than other phases (one-way ANOVA followed by Tukey's post test, p<0.05)."
Specific Comments: Line 213: I am not sure what is meant by 'important' here, consider clarifying or removing.

Response: We agree and "important" was removed and the sentence now reads, Line 231: "showed a shift towards the center of the sediment core".

Line 214-216: Given that a much better separation is found between d15N and OC source (Figure 6), why not regress d15N against the instantaneous C accumulation rate as you did for d13C in Figure 7?

Response: We agree and $\delta^{15}N$ against OC accumulation panel has been added to this Figure 6.

Line 232-235: This bit about data processing should likely be in the methods.

Response: We agree and have added the follow to Methods section (Line 161): "Organic carbon accumulation rates were estimated from an average between these the two dating methods, $^{239+240}Pu$ and $^{210}Pb_{ex}$ the dry bulk density (g cm$^{-3}$) and carbon content for each interval of the entire sediment core."

Line 259: Do you have evidence of the 'silting up' of this lake? Perhaps in the form of mass accumulation rates rather than OC, or, alternatively, looking for changes in DBD with depth.

Response: Mass accumulation rates, directly related to changes in DBD with depth, indicate a change in sediment source. However, this lake is situated in a region relatively preserved in the Amazon Forest. We now include the following, Line 297: "However, the lake is in a region relatively preserved, and therefore there is no other explanation other than deforestation in the margins to have caused the peak in OC burial between 1934 and 1975."

Line 264-272: An ANOVA (or similar approach) would be an appropriate test here to help put some weight behind this statement.

We agree and tested statistical differences (one-way ANOVA followed by the Tukey's post test) among phases of sedimentation. Se new Fig 7 and captions. We also have added information on the statistical analyse throughout the manuscript, including in a sentence added to the Results section (Line 234): " The significantly greater $\delta^{13}C$ peak and lower $\delta^{15}N$ values coupled to higher OC burial rates were observed in the phase between 1934-1975 in Jupindá Lake (one-way ANOVA followed by Tukey's post test, p<0.05; Fig. 7). The $\delta^{13}C$ values were greater in the phase 1934-1975 in relation to those previous and after respectively (one-way ANOVA followed by Tukey's post test, p<0.05)."

Figures 6 and 7: The C accumulation rates should probably be g/m2/yr, right?

The Reviewer is right and these corrections have been made to Figures 6 and 7 as suggested.

**Reviewer 2.**

This study investigates changes in organic carbon burial in an Amazonian floodplain lake in response to deforestation and urban development. The authors conclude that OC burial was strongly affected by initial deforestation and development close to the river and lake margins, but not very strongly during later deforestation which took place further away from surface waters.

We appreciate the Reviewers comments. Below we respond to each comment individually.

OC burial in lakes in an important but understudied carbon flux, and as such, studies like this are needed, particularly if coming from an understudied area of the world. Also, methodologically this study seems sound, and the conclusions seem plausible and supported have, however, one major concern regarding the dating methods and the subsequent calculation of OC burial rates. The authors use two methods, measurement of plutonium (Pu) from atmospheric fallout after nuclear bomb testing, and measurement of 210Pb which is a naturally occurring atmospheric radioisotope. The Pu method returns a sediment depth corresponding to pre-bomb testing ( ~1950), and therefore an average sediment accumulation rate since this time. The Pb method, on the other hand, can resolve changes in sediment accumulation rate over time, if combined with an appropriate model (e.g. the CRS model). Since OC burial is calculated as sediment accumulation rate * OC content * dry bulk density, the authors should use the Pb data. Instead, they calculate an average long-term mean sediment accumulation from the Pu and Pb data, and use this fixed, constant value for calculation of OC burial over time. It is not correct to present data on the temporal change of OC burial rates (Figs 6 and 7) that are based on keeping the most important term (sediment accumulation rate) constant. I would like to encourage the authors to recalculate the OC burial values based on the Pb chronologies, and also to present these Pb chronologies in a graph. Alternatively, they should provide arguments as to why their approach is warranted.

Response: We have added a new paragraph to discuss our dating approach, Line 222: Because the $^{210}Pb_{ex}$ activities are relatively uniform from the surface to the ~20 cm depth, the short-term measurements are not possible. However, from the $^{239+240}Pu$ data, one can say with certainty that the material below 22 cm was deposited pre-bomb (that is, prior to the early 1950s). This affixes an upper limit on the sedimentation accumulation rate (SAR), form 1950 to 2010, to be near 3.8 cm/year. This accretion rate is similar to the $^{210}Pb$ rates and we therefore conclude that the sedimentation rates have not changed significantly during previous ~120 years.

In the following, I provide some more detailed comments.

87-101 This is the site description, and it should be moved to methods. Instead, elaborate here more about the ways you think deforestation and exploitation can affect OC burial in lakes, and mention and cite previous findings on this topic explicitly. Also, it would be good to see a clearly formulated, testable hypothesis spelled out in the Introduction.

Response: This paragraph has been moved to the methods as suggested and the manuscript now includes a hypothesis, Line 91: "We hypothesize that the well documented records on

historical deforestation in this region of the Amazon Basin is related to the carbon burial capacity of the floodplain lakes."

L115. How was the acidification done? There are papers showing that different acidification methods affect the stable isotopic signature differently.
Response: we now include on the acidification, Line 116: "10% HCl following the procedures outlined in Naidu et al. (2000)". This procedure is routinely run in our laboratory. To further demonstrate the precision of the methodology we have added the following , Line 119-122: "Working standards were used (glucose, 10.7 ppt and urea, 41.3 ppt) to calibrate for d13C. A pair of standards were measured with every 20 samples. These standards were calibrated initially against international absolute standards LSVEC and NIST8542."

L 120. What was the motivation to use both Pu and Pb dating methods? Why not Pb alone? It seems that the Pu method is not suitable for resolving changes in sediment. accumulation rate over time, which is the purpose of this study. Is the sole purpose of the Pu method to derive an independent estimate of long-term average accumulation?
Response: Correct, the Pu is used here to substantiate the $^{210}$Pb dating method. We feel our sediment dating is more confident using two independent methods.

L194. Fig 43 must be a typo.
Correction made.

L204-206. Unclear what this means, and what the equation describes.
Response: This sentence discusses the bioturbation up to the 20 cm depth and the equation shows a $LN^{210}Pb_{ex}$ vs Depth linear regression below the mixed layer. We have added the following to the manuscript, Line 215: "This indicates that the sedimentation is constant below the 20 cm depth."

L217. Were the OC burial rates reported in the paragraph calculated using the CIC or CRS model on 210Pb data?
Response: An average was taken between the two dating methods, i.e. 4.2 mm yr$^{-1}$, to obtain a $^{210}$Pb sedimentation rate. The following has been added to the manuscript (Line 218): "In order to obtain a more reliable estimates of the historical carbon burial rates, an average was taken between these the two dating methods, $^{239+240}$Pu and $^{210}$Pb$_{ex}$ (4 mm year$^{-1}$), and multiplied by the DBD and OC content for each interval of the entire sediment core."

L221-225. This sentence is difficult to understand and long. Rephrase, please.
Response: Change made and now reads (Line 245), "In relation to land use/cover in the surrounding fluvial channels and the Jupindá lake, only the smallest buffer (100 m) showed significant changes during 1934-1975. This time period is when deforestation was nearly 75% higher than in the subsequent time period 1975-2008 (Figure 8a) and when OC burial was greatest ((Figure 8b)."

L228. I do not understand this reasoning. You have calculated 210Pb chronologies, and the activity profile (Fig.4) does show indications of changes in sediment accumulation rate over time. So why do you use an average long-term accumulation rate, based on the Pb method and the Pu method (which does not give any resolution in time of accumulation rate)? Why don't you trust the Pb chronologies? Using an aver- age long-term accumulation rate is contrary to the purpose of your study. Here's also a conceptual flaw: if you present changes over time in OC burial rate (Figs 6 and 7), which do not account for changes over time in sediment accumulation rate, then these graphs only reflect changes in OC content and dry bulk density.

See response above.

L264-267. It is difficult to see this linkage between the buffer zones, time periods and OC burial rates. Can you think of a way to illustrate these links in a graph?
A new panel to Figure 8 has been added to better show the link between buffer zones, periods and OC burial rates.

L276-278. This is an important conclusion.
We appreciate this positive feedback.

L283-284. But only concerning long-term average sediment accumulation rates, while the purpose of this study was to investigate temporal changes of sediment accumulation rate, which can only be accomplished with the Pb data. This distinction should be made throughout the ms, and the Pb chronologies using both the CIC and CRS models should be shown.
See responses above the on the the similar sedimentation rates for different time periods based on the two independent sediment dating methods presented in this work.

Figs 6 and 7. The axis label should be "OC burial rate", not just "OC"
Changes made as suggested

Reference
Naidu, A. S., L. W. Cooper, B. P. Finney, R. W. Macdonald, C. Alexander, and I. P. Semiletov. 2000. Organic carbon isotope ratio ($\delta^{13}$C) of Arctic Amerasian Continental shelf sediments. International Journal of Earth Sciences **89**:522-532.

Once again, we thank the Reviewers for these constructive comments. We feel that these comments significantly strengthened our manuscript.

END

---

## Author Response (AR2)

**Response to Reviewer's comments for the manuscript bg-2017-151:**

**"Historic carbon burial spike in an Amazon floodplain lake linked to riparian deforestation near Santarem, Brazil"**

Note that the line numbers refer to the revised manuscript.

Editor Comments to the Author:

Your revised version has now been re-evaluated by both original reviewers. As you will see from their comments, both agree that the manuscript has been substantially improved but they also raise a few minor additional comments which should be straightforward to address (Referee # 1) or come back to comments/suggestions that were not adequately addressed (Referee #2, issue of Pu & Pb methodologies to quantify sedimentation rates for both periods considered). I would therefore like to encourage you to revise your manuscript, with particular attention to the issue of sedimentation rates mentioned by Reviewer#2, and provide a detailed response explaining how this is addressed in a new version of the ms.

**RESPONSE:** We appreciate these construction comments. Below we respond to each comment individually. Note that we have incorporated many of the suggestions on the issue of sedimentation rates as recommended by Reviewer#2. We feel these modifications have strengthened the manuscript substantially.

Reviewer 1

General Comments:

Sanders et al. have revised their manuscript on the organic matter accumulation rates within a floodplain lake in the Amazon. This revised manuscript presents an improved analysis of the data and presentation of methods. In addition to a few small comments below, I have one major question: does any of this impact how these systems should be managed? Currently, neither the abstract nor the conclusion tackle this question. It may not be completely answerable within the scope of this study, but I would welcome an attempt at question of land use management. I believe this would improve the quality of the manuscript, especially as there is significant text spent analyzing the land use change component of this study system.

**RESPONSE:** We agree and have added the following sentences.

In the abstract:

(Lines 51-53) - "Therefore, this supports the conservation priority of riparian forests as an important management practice for Amazon flooded areas."

In the conclusion:

(Lines 234-235) - "However, any increase of OC burial rates attributed to deforestation might be lower than that loss of terrestrial biomass in the standing crop or soils."

(Lines 328-329) - "This work supports the urgent need for management practices based on the conservation of riparian forests, demonstrating the spatial dependence of carbon burial capacity of the Amazon floodplain lakes with respect to advances in deforestation in the Basin."

Specific Comments:

Lines 115-120: I did not catch this in my initial review, but the d15N results should be interpreted with caution based on this pre-treatment method. The relative differences between samples is possibly retained, but the absolute value of d15N following acid pre-treatment is suspect. In addition, C:N can also be affected as acidification and should be interpreted with caution. I do not believe this invalidates the findings here, however these caveats should be considered when interpreting the data.
Here are two recent references that document these issues:

Brodie, C.R., Casford, J.S.L., Lloyd, J.M., Leng, M.J., Heaton, T.H.E., Kendrick, C.P., Yongqiang, Z., 2011. Evidence for bias in C/N, $\delta13C$ and $\delta15N$ values of bulk organic matter, and on environmental interpretation, from a lake sedimentary sequence by pre-analysis acid treatment methods. Quaternary Science Reviews 30, 3076–3087.

Kim, M.S., Lee, W.S., Suresh Kumar, K., Shin, K.H., Robarge, W., Kim, M., Lee, S.R., 2016. Effects of HCl pretreatment, drying, and storage on the stable isotope ratios of soil and sediment samples. Rapid Communications in Mass Spectrometry 1567–1575.

RESPONSE: We agree and have added the following, Line 121-122: "The $\delta^{15}N$ results and the C/N ratios results should be interpreted with caution based on this pre-treatment method (Brodie et al. 2011)."

Lines 270-283: This fleshed out handling of the statistical methods if welcome. However, I wonder if some of this detail is not better fit for the methods and/or results sections so that the discussion can focus on the implications of the findings.

RESPONSE: We agree and moved these phrases to the Methods section (Line 197 - 204).

Figure 7: Please consider adding a secondary Y-axis indicating the depth associated with the year.

RESPONSE: We agree and the Y-axis to Figure 7 now contains the depth instead of year as also suggested by Reviewer 2.

**Reviewer 2**

I have reviewed the first version of this paper, and in my first review, I had concerns about the dating methods. Unfortunately, I do not find that the authors have addressed my concerns appropriately in their revision.
**RESPONSE:** Reviewer 2 raises some valid points in this review of which we feel has improved the manuscript substantially. We detail how we have changed the manuscript in accordance to this Reviewer's suggestions at the end of this comment, as it all relates to the same topic.

The problem is the following: two different dating methods are used, the Pu method and the Pb method.

The Pu method gives information about the average sediment accumulation rate since ~1950, but does not allow for finer temporal resolution. The authors use it in that sense, and that's fine. AD 1950 was located at ~20 cm depth.

The Pb method can resolve temporal changes in sediment accumulation rate at the scale of years, but in the study lake, the authors state that the Pb profile was disturbed in the upper 20 cm, but declining linearly in the 20-60 layer layer. Therefore, they did not calculate sediment accumulation from the Pb data in upper 20 cm of sediment, and they present an average sediment accumulation rate of 4 mm/yr in the 20-60 cm layer.

The chronology given in this paper is therefore (a) an average sediment accumulation rate since 1950 (0-20 cm sediment depth) from the Pu dating, and (b) an average sediment accumulation rate prior to 1950 (20-60 cm sediment depth) from the Pb dating. The presented data does not give information about how sediment accumulation might have varied during the 1950-present period, or during the ~1850-1950 period. Contrary to what the authors claim (L225-227), the fact that the average sediment accumulation rates for these two periods are similar does not indicate that there was little change of sediment accumulation over time; within each of the time periods, sediment accumulation rates might varied at the scale of years. In fact, when looking at the 210Pb(ex) profile (Fig 4b), there is an intriguing increase in unsupported Pb over the 3-20 cm horizon, possibly indicating non-constant sediment accumulation. Similarly, 210Pb(ex) varies between layers in the 20-60 cm horizon, indicating quite variable sediment accumulation rate, and not a constant sediment accumulation rate, as the authors claim (L212-215).

The lack of time-resolved chronologies is serious when the authors plot chemical data over age (Fig. 7), since there is no information about the exact age each respective layer of sediment. In Figure 7, the authors should instead plot chemical data against sediment depth, and indicate the AD 1950 layer.

The same problem appears when changes in OC burial are attributed to changes in land use (Fig.8B). Over the period where land use change was most intense, e.g. the past 50 years, the authors have only one average value of sediment accumulation rate to rely on, and the values of OC burial cannot be confidently attributed to a certain year. In fact, the calculations of OC burial use one average sediment accumulation rate, and using such averaged OC burial rates to illustrate temporal changes in burial seems unwarranted. Analysis of temporal patterns can only be done with temporally resolved data.

In my opinion, the authors can go two ways: either present the Pb chronologies (i.e. plot depth against age, based on both CIC and CRS models) in order to present evidence for the attribution of individual sediment layers to certain years. I asked for the Pb chronologies in my first review, but the authors have not responded to that. The other potential way to deal with this issue would be to stay away from giving distinct ages to distinct sediment layers, and analyze the differences between pre-1950 and post-1950 layers (for which there is good data from the Pu dating). The average sediment accumulation rates could still be used to speak about indicative ages, i.e. since the long-term average sediment accumulation was ~4 mm/yr, the 0-4 cm layer may be regarded to represent approximately the past 10 yrs.

Whichever way the authors choose, the ms needs to be revised accordingly prior to publication.

**RESPONSE:** We have decided to focus the manuscript on general range approximations of the indicative ages, instead of distinct ages to distinct sediment layers as recommended by this Reviewer, i.e., >1930, 1930-1970 and 1970-2010, on Figures 7 and 8A and throughout the manuscript. For instance, Line 39-41: "Historical records from the 1930s and satellite data from the 1970s were used to calculate deforestation rates between 1930 and 1970, and 1970 to 2010 in four zones". We have also changed Figures 7 and 8 to show depths instead of specific year as suggested. We feel these general age ranges best reflects the uncertainties associated with our sediment dating methods, as noted by this Reviewer.

Furthermore, we now use pre-1950 and post-1950 sedimentation rate estimates for the two separate layers as suggested by the Reviewer, i.e. The $^{239+240}$Pu for the rates from near 1950 to present (3.8 mm year$^{-1}$) and from ~1890 to approximately the 1950s (4.2 mm year$^{-1}$) as calculated from the $^{210}$Pb$_{ex}$ profiles. These rates for each sediment depth were multiplied by the DBD and OC content for each interval along the entire sediment core.

Another comment: I found the hypothesis a bit vaguely formulated, "related to the carbon burial capacity" leaves room for interpretation. The authors could be more definitive in their prediction: "historically documented increases in deforestation have increased OC burial rates in the studied floodplain lake" or something similar.

**RESPONSE:** We agree and have changed the hypothesis to the following, Line 95 - 96: "
[revised manuscript text omitted]

[Figure]

- Curuá-Una River
- Floodplain Lake Jupindá

 **Figure 2.**

[Figure]

**Figure 3.**

[Figure]

**Figure 4.**

[Figure]

**Figure 5.**

[Figure]

**Figure 6.**

[Figure]

**700** **Figure 7.**

[Figure]

**701**
**702**
**703**
**704**

[Figure]

**705**
**706**
**707**
**708**
**709**
**710**
**711**
**712**

**Figure 8.**

[Figure]

[Figure]

---

## Author Response (AR3)

Dear Bouillon,

Thank you for your positive reply. Below is a response to your minor suggestion. We have also uploaded the final version of this manuscript along with all of the original figures.

A final minor suggestion: In response to Ref#1, you added a sentence mentioning that C/N ratios and d15N should be interpreted with caution "based on this pre-treatment method", I suggest you specify what type of bias (magnitude, direction) has been reported in the paper you cite.

[revised manuscript text omitted]

- Curuá-Una River
- Floodplain Lake Jupindá

**Figure 2.**

[Figure]

**Figure 3.**

[Figure]

**Figure 4.**

[Figure]

**Figure 5.**

[Figure]

**Figure 6.**

[Figure]

**Figure 7.**

[Figure]

**Figure 8.**

[Figure]

[Figure]